# CellPLM: Pre-training of Cell Language Model Beyond Single Cells

## Abstract

The current state-of-the-art single-cell pre-trained models are greatly inspired by the success of large language models. They trained transformers by treating genes as tokens and cells as sentences. However, three fundamental differences between single-cell data and natural language data are overlooked: (1) scRNA-seq data are presented as bag-of-genes instead of sequences of RNAs; (2) Cell-cell relations are more intricate and important than inter-sentence relations; and (3) The quantity of single-cell data is considerably inferior to text data, and they are very noisy. In light of these characteristics, we propose a new pre-trained model *CellPLM*, which takes cells as tokens and tissues as sentences. In addition, we leverage spatially-resolved transcriptomic data in pre-training to facilitate learning cell-cell relationships and introduce a Gaussian mixture prior distribution as an additional inductive bias to overcome data limitation. *CellPLM* is the first single-cell pre-trained transformer that encodes cell-cell relations and it achieves state-of-the-art performance in various downstream tasks.

## 1 Introduction

Next-generation sequencing technologies such as single-cell RNA sequencing (scRNA-seq [1]) have produced vast amounts of data, sparking a surge of interest in developing large-scale pre-trained models for single-cell analysis [2, 3, 4, 5]. These models seek to capture underlying structures and patterns from unlabeled scRNA-seq data, and can be fine-tuned on specific downstream datasets to deliver accurate predictions and nuanced insights into cellular mechanisms. Particularly, these pre-trained models have been inspired by the success of large language models, such as BERT and GPT [6, 7], and treat genes as words (tokens) and cells as sentences to train transformers [8]. However, we argue that these approaches may have limitations due to the fundamental differences between single-cell data and natural language data, which have been largely overlooked in existing literature:

*First*, unlike sentences, the scRNA-seq data utilized by existing pre-trained models are not sequential. Before the training stage, RNA sequences have been identified as functional units, i.e., genes. Instead of original sequences, data is denoted as a cell-by-gene count matrix that measures the abundance of individual genes within each cell. This is analogous to bag-of-words model in natural languages, where the set of genes is fixed, and there is no sequential relationship among them.

*Second*, the relationship between cells is remarkably more intricate and important than that of sentences, since cell-cell interactions play an essential role in determining cell states and cell development [9]. Additionally, within tissues, there are numerous cells from the same or similar cell lineage, which grants them similar gene expression profile and hence provides valuable supplementary information for denoising and identifying cell states [10, 11, 12]. As a result, many recent methods [13, 14, 15, 16] have constructed cell-cell graphs to advance representation learning for single-cell data.

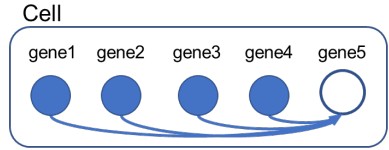
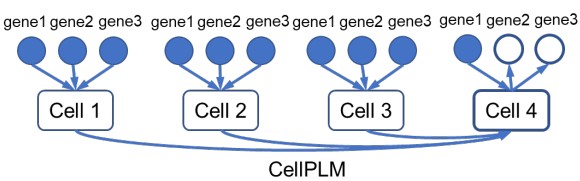

Figure 1: An illustration of the difference in the language models between existing single-cell pre-trained models and *CellPLM*. Existing pre-trained models only consider conditional probability between gene expressions within the same cell, while in *CellPLM*, gene expression distribution is also conditioned on other cells. See details in Section 3.

Such evidence demonstrates the importance of cell-cell relationship, which is usually neglected by existing pre-trained models.

*Third*, the quantity and quality of single-cell datasets are significantly lower than those of natural language data. For comparison, the high-quality filtered English dataset extracted from Common Crawl corpora [17] consists of 32 billion sentences, whereas the largest collection of single-cell datasets, namely the Human Cell Atlas [18], includes less than 50 million cells. To make things worse, single-cell data often suffer from technical artifacts and dropout events [19, 20], as well as significant batch effects between sequencing platforms and experiments [21, 22].

The aforementioned differences introduce distinct challenges which call for new pre-training strategies tailored for single-cell data. To bridge this gap, we propose a novel single-**Cell P**re-trained **L**anguage **M**odel (*CellPLM*), which addresses these challenges from following perspective: **First**, As shown in Figure 1, *CellPLM* proposes a cell language model to account for cell-cell relations. The cell embeddings are initialized by aggregating gene embeddings since gene expressions are bag-of-word features. **Second**, *CellPLM* leverages a new type of data, spatially-resolved transcriptomic (SRT) data, to gain an additional reference for uncovering cell-cell interactions. Compared to scRNA-seq data, SRT data provide additional positional information for cells. Both types of data are jointly modeled by transformers. **Third**, *CellPLM* introduces inductive bias to overcome the limitation of data quantity and quality by utilizing a Gaussian mixture model as the prior distribution in the latent space. This design can lead to smoother and better cell latent representations [23, 15, 24]. To the best of our knowledge, the proposed *CellPLM* is the first pre-trained transformer framework that encodes inter-cell relations, leverages spatially-resolved transcriptomic data, and adopts a reasonable prior distribution. It is evident from our experiments that *CellPLM* demonstrates superior performance in various downstream tasks.

## 2 Single-cell Pre-trained Models

Deep learning methods for single-cell data have garnered significant research interest in recent years [11]. However, due to the distinct model architectures, the knowledge learned by models is not transferable across tasks. To address this issue, there is an emerging effort [2, 3, 4, 5] from the research community to explore the potential of a foundation model that first extracts latent knowledge from unlabeled scRNA-seq data and subsequently generalizes this knowledge to a variety of tasks.

The first such pre-trained model for single-cell data, scBERT [2], takes genes as tokens and leverages an efficient transformer [25] to encode over 16,000 gene tokens for each cell. By randomly masking a fraction of non-zero gene expression values and predicting them based on the remaining data, scBERT effectively learns intricate relationships between genes, leading to improved cellular representation. Later, xTrimoGene [3] made two key enhancements to scBERT: pruning zero-expressed genes and improving expression binning strategies by an auto-discretization strategy. These modifications notably enhance scalability and feature resolutions. Another latest preprint, scGPT [5], introduces a variant of masked language modeling that mimics the auto-regressive generation in natural language processing, where the masked genes are iteratively predicted according to model's confidence. Unlike the aforementioned models, tGPT [4] completely abandons masked language modeling. It constructs sequences of genes based on the ranking of gene expressions within each cell, and the model is trained to autoregressively predict the name of the next gene. Despite discarding the precise expressions,

this approach demonstrates enhanced robustness against batch effects and can be generalized to bulk RNA data.

The aforementioned models all regard genes as tokens and focus solely on modeling gene relationships within individual cells, neglecting the intercellular information in an organism. In contrast, *CellPLM* overcomes this limitation by introducing a cell language model that extends beyond single cells. Furthermore, by leveraging the spatial information of cells acquired from SRT data, along with a prior Gaussian mixture distribution, the model achieves unparalleled performance on a range of downstream tasks.

# 3 Cell Language Model Beyond Single Cells

In this section, we introduce the concept of the cell language models and detailed implementation of the proposed *CellPLM*. As illustrated in Figure 2, *CellPLM* consists of four modules: a gene expression embedder, an encoder, latent space, and a decoder, which we will demonstrate in Section 3.2. At a higher level, there are two stages in our framework: pre-training and fine-tuning. During pre-training, the model is trained on unlabeled data with a masked language modeling objective. For fine-tuning, the model is first initialized with the pre-trained parameters, and then all of the parameters are fine-tuned using data and labels (if available) from the downstream datasets. We demonstrate the pre-training and fine-tuning framework in Section 3.3 and 3.3, respectively.

## 3.1 Cell Language Model

Due to the recent achievements of large language models [7], several studies [2, 3, 4, 5] have drawn inspiration from natural language processing in an attempt to establish a foundational model for single-cell analysis. These studies consider genes as tokens and train transformers on them, aiming to model the conditional probability between gene expressions. Concretely, previous pre-trained models are trained on scRNA-seq data, which are stored in the format of a cell-by-gene matrix $\mathbf{X} \in \mathcal{R}^{N \times k}$, where $N$ is the number of cells, and $k$ is the number of distinct gene types. The value of $\mathbf{X}_{i,j}$ denotes the count of gene $j$ observed in cell $i$, also known as gene expression. The pre-training goal of these models is to estimate a conditional probability distribution, which can be formulated as:

$$p\left(\mathbf{X}_{i,j} | \{\mathbf{X}_{i,o}\}_{o \in \mathcal{O}(i)}\right), j \in \mathcal{U}(i), \tag{1}$$

where $i$ refers to the $i$-th cell and $\mathcal{O}(i)$ is the set of observed genes in cell $i$ whose expressions are known; $\mathcal{U}(i)$ denotes the set of unobserved genes in cell $i$ whose expression will be predicted by the model, typically referring as masked genes. If we consider genes as words, this objective is analogous to the language model in computational linguistics [26], and thus can be named a "gene language model". In this way, the model is trained to capture the intrinsic relations between genes, which can provide prior knowledge for downstream analysis.

However, in Eq. (1), the distribution of unobserved gene expressions only depends on genes within the same cell, while disregarding the information of other cells within the same tissue, which does not align with the inherent nature of biology. Therefore, in *CellPLM*, we provide a different perspective to model scRNA-seq data by treating cells as tokens:

$$p\left(\mathbf{X}_{i,j} | \{\mathbf{X}_{u,v}\}_{(u,v) \in \mathcal{M}^C}\right), (i,j) \in \mathcal{M}, \tag{2}$$

where we denote $\mathcal{M}$ as the set of masked gene expressions in $\mathbf{X}$, and $\mathcal{M}^C$ is the complement, i.e., the set of unmasked expressions. The distribution of a masked entry $\mathbf{X}_{i,j}$ depends on both the observed genes in cell $i$ and genes from other cells that are not masked. We hereby name it as "cell language model", which models the distribution of cellular features beyond single cells. By estimating the conditional probability distribution in Eq. (2), *CellPLM* is trained to capture the intricate relationships that exist between not only genes but also cells.

From a biology perspective, there are particularly two types of inter-cell relations that can be beneficial to *CellPLM*. First, within tissues, there are numerous cells from the same or similar cell lineage, which mutually provide valuable supplementary information for denoising and identifying cell states [10, 11, 12]. The other type of relations, cell-cell interactions (a.k.a, cell-cell communications), plays an essential role in determining cell development and cell states [9]. Existing analysis methods [27, 28, 29] have already explored the cell-cell communications on the cell type or cluster levels, while

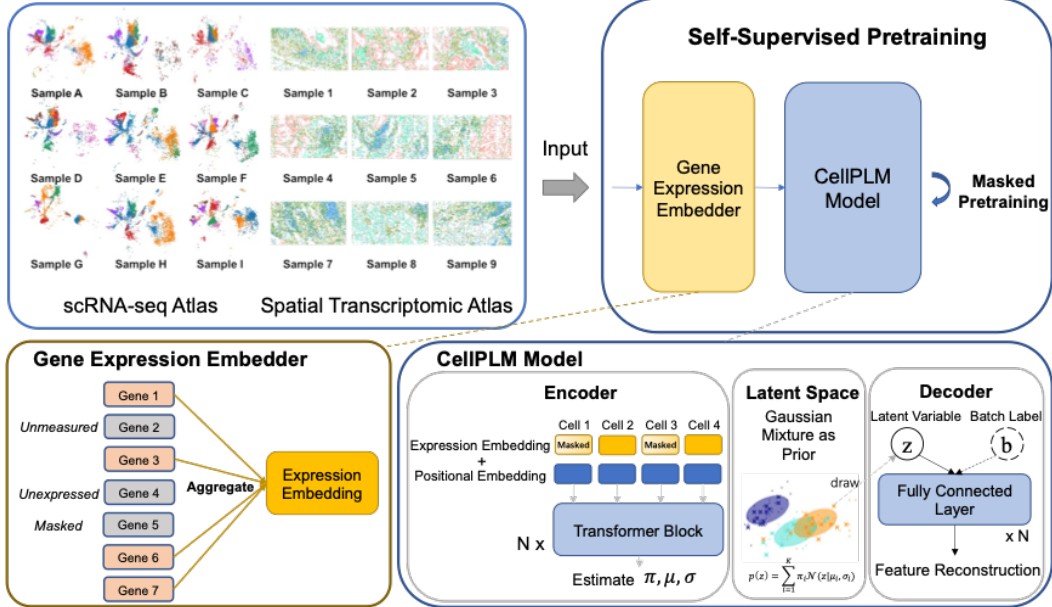

Figure 2: An illustration of the pre-training framework of *CellPLM*. *CellPLM* is pre-trained with cell-level masked language modeling task. The model consists of four modules: a gene expression embedder, a transformer encoder, a gaussian mixture latent space, and a batch-aware decoder.

*CellPLM* aims to capture the intricate "language" of cell-cell communications between single cells. Overall, *CellPLM* presents a novel cell language model that aligns well with biological principles and holds great potentials to enhance downstream tasks by extracting valuable cellular knowledge from unlabeled single-cell data.

## 3.2 Model Architecture

**Gene Expression Embedder**. The first module in *CellPLM* model is a gene expression embedder, which projects input gene expressions into a low-dimensional cellular feature space. In light of the nature that scRNA-seq is profiled as bag-of-genes features, *CellPLM* learns an embedding vector for each type of gene, and then aggregates these gene embeddings according to their expression levels in each cell. Formally speaking, for gene $j \in \{1, ..., k\}$, a learnable embedding vector $\mathbf{h}_j \in \mathcal{R}^d$ is assigned, where $d$ is the hidden dimension of the encoder layers. $\mathbf{h}_j$ can be either randomly initialized or initialized by prior knowledge, e.g., gene2vec [30]. The gene expression embedding matrix $\mathbf{E} \in \mathcal{R}^{N \times d}$ is then generated by aggregating gene embeddings according to their expressions:

$$\mathbf{E}_i = \sum_{j=1}^{k} \mathbf{X}_{i,j} \mathbf{h}_j, \tag{3}$$

where $\mathbf{E}_i$ is the $i$-th row vector of $\mathbf{E}$, corresponding to the gene expression embedding for cell $i$. Note that the gene expression matrix $\mathbf{X}$ is a sparse matrix since the zero-rate of scRNA-seq can be up to 90% [31]. In addition, unmeasured genes (per sequencing platforms) also lead to zero entries in $\mathbf{X}$. Therefore, when implementing Eq. (3), *CellPLM* leverages a sparse linear layer instead of a regular fully connected layer. This significantly improves memory and computational efficiency.

**Transformer Encoder**. The proposed *CellPLM* follows an encoder-decoder structure, where the encoder is based on transformers [8]. The transformer model was originally developed for processing textual data. It leverages multi-head self-attention mechanisms to capture relationships between input tokens and incorporates positional encoding to represent the token positions. In *CellPLM*, by considering cells as tokens, we can readily apply the transformer model to capture intercellular relationships. When applying the transformer, we consider the embedding at $l$-th layer $\mathbf{H}^{(l)} \in \mathcal{R}^{N \times d}$ as a set of $N$ tokens, where $N$ is the total number of cells in a tissue sample, and $d$ is the hidden dimension. By stacking $L$ transformer layers, *CellPLM* gradually encodes cellular and inter-cellular

information into cell embeddings, formulated as:

$$\mathbf{H}^{(l)} = \text{TransformerLayer}^{(l)}(\mathbf{H}^{(l-1)}). \tag{4}$$

In practice, $N$ can scale up to ten thousands, which is out of the capacity of an ordinary transformer. Therefore, we adopt an efficient variant of transformers with linear complexity (i.e., Performer [25]) for the implementation of transformer layers.

To further inform inter-cellular relations, we incorporate spatial positional information of individual cells from a novel type of data, spatially-resolved transcriptomic (SRT) data. Specifically, SRT data consist of two parts. One is a gene expression matrix $\mathbf{X} \in \mathcal{R}^{N \times k}$ same as scRNA-seq data, and the other part is a 2D coordinate matrix $\mathbf{C} \in \mathcal{R}^{N \times 2}$. The coordinates denote the center position of each cell within a field-of-view (FOV) where the cells are located (an illustration can be found in Appendix A). This feature helps locate the microenvironment surrounding each cell, providing an additional reference for identifying cell lineage and cell communications, which were introduced in Section 3.1. To encode this extra positional information, we leverage the idea of positional encodings (PE) in transformers. Since sinusoidal PE achieves competitive performance and has lower complexity on SRT data [16], we generate a 2D sinusoid PE for cells in SRT data, denoted as $\mathbf{P} \in \mathcal{R}^{N \times d}$, where $\mathbf{P}_i$ is the $d$ dimensional PE vector for cell $i$ (see details in Appendix B). For scRNA-seq data, a randomly initialized $d$-dimensional vector $p'$ is shared among all cells, which also results in a placeholder PE matrix $\mathbf{P}$. The initial cell embeddings are now formulated as $\mathbf{H}^{(0)} = \mathbf{E} + \mathbf{P}$, where $\mathbf{E}$ is the expression embeddings from Eq. (3) and $\mathbf{P}$ is the positional embeddings.

**Gaussian Mixture Latent Space**. One of the highlights of *CellPLM* is the design of probabilistic latent space. Prior studies have employed variational autoencoders for single-cell analysis, which typically assumes an isotropic Gaussian distribution as the prior distribution of the latent space [32, 33]. While this approach can effectively remove batch effects, it may also result in a loss of information regarding the underlying biological structure of cell groups. To address this limitation, *CellPLM* incorporates the concept of Gaussian mixture variational encoder [34, 35, 15], which utilizes a mixture of Gaussians to capture the information of distinct functional groups of cells. Formally, for $i \in \{1, \ldots, N\}$, the generative model of cell $i$ can be formulated as:

$$
\begin{aligned}
p(\mathbf{y}_i; \boldsymbol{\pi}) &= \text{Multinomial}(\boldsymbol{\pi}), \\
p(\mathbf{z}_i \mid \mathbf{y}_i) &= \prod_{i=1}^{L} \mathcal{N}\left(\boldsymbol{\mu}_{y_{i,l}}, \text{diag}\left(\boldsymbol{\sigma}_{y_{i,l}}^2\right)\right), \\
p_{\theta_{dec}}(\mathbf{x}_i \mid \mathbf{z}_i) &= \mathcal{N}\left(\boldsymbol{\mu}_{\mathbf{z}_i}, \sigma^2 \mathbf{I}\right),
\end{aligned}
\tag{5}
$$

where $\mathbf{y}_i \in \mathcal{R}^L$ represents the one-hot latent cluster variable and $\boldsymbol{\pi}$ is its prior; $y_{i,l}$ denotes the $l$-th entry of $\mathbf{y}_i$; $\boldsymbol{\mu}_{y_l} \in \mathcal{R}^{d_z}$ and $\boldsymbol{\sigma}_{y_l}^2 \in \mathcal{R}^{d_z \times d_z}$ denote the mean and variance of the $l$-th Gaussian component, respectively; and $\boldsymbol{\mu}_{z_i} \in \mathcal{R}^k$ and $\sigma^2 \mathbf{I} \in \mathcal{R}^{k \times k}$ denote the posterior mean and variance of expression $\mathbf{x}_i$, respectively. In this work, we assume that $\sigma^2$ is a constant and the posterior mean is parameterized by $\boldsymbol{\mu}_{z_i} = f_{dec}(\mathbf{z}_i; \theta_{dec})$.

To estimate the posterior of $\mathbf{z}_i$ and $\mathbf{y}_i$, we parameterize the inference process with neural networks. Specifically, we assume that the cluster variables $\mathbf{y}$ are independent of the expression $\mathbf{x}_i$ condition on latent variables $\mathbf{z}_i$. The inference model can be formulated as:

$$
\begin{aligned}
q_{\eta_\mu, \eta_\sigma}(\mathbf{z}_i \mid \mathbf{x}_i) &= \mathcal{N}\left(\hat{\boldsymbol{\mu}}_i, \text{diag}\left(\hat{\boldsymbol{\sigma}}_i^2\right)\right), \\
q_{\eta_\pi}(\mathbf{y}_i \mid \mathbf{z}_i) &= \text{Multinomial}(\hat{\boldsymbol{\pi}}_i),
\end{aligned}
\tag{6}
$$

where the estimations are given by

$$
\begin{aligned}
\mathbf{h}_i &= f_{enc}(\mathbf{x}_i; \eta_{enc}), \\
\hat{\boldsymbol{\mu}}_i &= f_\mu(\mathbf{h}_i; \eta_\mu), \\
\log\left(\hat{\boldsymbol{\sigma}}_i^2\right) &= f_\sigma(\mathbf{h}_i; \eta_\sigma), \\
\hat{\boldsymbol{\pi}}_i &= f_\pi(\mathbf{z}_i; \eta_\pi).
\end{aligned}
\tag{7}
$$

Here $f_{enc}(\cdot; \eta_{enc})$ represents the transformer encoder, $f_\mu(\cdot; \eta_\mu)$, $f_\sigma(\cdot; \eta_\sigma)$ and $f_\pi(\cdot; \eta_\pi)$ are neural networks. A log-evidence lower bound (ELBO) can be derived from this generative model for the optimization purpose [34]. However, as mentioned in Section 3.1, our pre-training framework

incorporates a cell language model, where parts of the input gene expression matrix $\mathbf{X}$ are masked. This will result in a modified objective. To formalize the problem, recall that previously we defined the masked set as $\mathcal{M}$. On top of that, we denote $\mathbf{M} \in \mathcal{R}^{N \times k}$ as a mask indicator matrix such that

$$\mathbf{M}_{i,j} = \left\{ \begin{array}{ll} 1 & \text{if } (i,j) \notin \mathcal{M}, \\ 0 & \text{if } (i,j) \in \mathcal{M}. \end{array} \right.$$

Let $\tilde{\mathbf{X}} \in \mathcal{R}^{N \times k}$ be the masked gene expression matrix given by the element-wise multiplication $\tilde{\mathbf{X}} = \mathbf{M} \odot \mathbf{X}$. The objective of cell language model with Gaussian mixture prior, i.e., a denoising variational lower bound [36], can be formulated as:

$$
\begin{aligned}
\mathcal{L}_{\text{CellLM}} =& \mathbb{E}_{q(\mathbf{Z}, \mathbf{Y} | \tilde{\mathbf{X}})} \mathbb{E}_{p(\tilde{\mathbf{X}}|\mathbf{X})} \left[ \ln \frac{p_\theta(\mathbf{X}, \mathbf{Z}, \mathbf{Y})}{q_\eta(\mathbf{Z}, \mathbf{Y} | \tilde{\mathbf{X}})} \right] \\
=& \underbrace{\mathbb{E}_{q_{\eta_{enc}}(\mathbf{Z}|\tilde{\mathbf{X}})} \mathbb{E}_{p(\tilde{\mathbf{X}}|\mathbf{X})} \left[ \log p_{\theta_{dec}}(\mathbf{X} | \mathbf{Z}) \right]}_{\mathcal{L}_{\text{recon}}} - \underbrace{\mathbb{E}_{q_{\eta_\pi}(\mathbf{Y}|\mathbf{Z})} \left[ \text{KL} \left( q_{\eta_{enc}}(\mathbf{Z} | \tilde{\mathbf{X}}) \| p(\mathbf{Z} | \mathbf{Y}) \right) \right]}_{\mathcal{L}_{\text{cond}}} \\
& - \underbrace{\mathbb{E}_{q_{\eta_{enc}}(\mathbf{Z}|\tilde{\mathbf{X}})} \left[ \text{KL} \left( q_{\eta_\pi}(\mathbf{Y} | \mathbf{Z}) \| p(\mathbf{Y}) \right) \right]}_{\mathcal{L}_{\text{Y}}}.
\end{aligned}
$$
(8)

Similar to previous works [34], we refer to the three terms in Eq. (8) as reconstruction term $\mathcal{L}_{\text{recon}}$, conditional prior term $\mathcal{L}_{\text{cond}}$ and $\mathbf{Y}$ prior term $\mathcal{L}_{\text{Y}}$. The approximation and estimation of the denoising variational lower bound are specified in Section 3.3.

**Batch-aware Decoder**. The decoder in *CellPLM* operates by decoding each cell individually, given that the tissue context has already been encoded into the latent space by the encoder. The decoder's purpose is twofold: to reconstruct masked features and to help remove batch effects from the latent space. In order to accomplish this goal, the decoder stacks several feed-forward layers (FFLayers) atop the input of latent variables $\mathbf{z}$, and a batch embedding, denoted as $\mathbf{b} \in \mathcal{R}^{d_z}$. Specifically, for each cell, the batch embedding is loaded from a learnable lookup table as $\mathbf{b} = \text{LookUp}(b)$, where $b$ is the label indicating the specific tissue sample (or FOV for SRT data) from which the cell has been drawn. By feeding the batch label to the decoder, a batch-effect-free latent space can be achieved, as empirically evidenced in scVI [32]. The decoder can thus be formulated as:

$$\mathbf{h}^{(0)} = \mathbf{z} + \mathbf{b}, \quad \mathbf{h}^{(l)} = \text{FFLayer}^{(l)}(\mathbf{h}^{(l-1)}),$$

where $l$ indicates the number of the layer, $\mathbf{h}^{(l)}$ is the hidden vector of layer $l \in (1..L-1)$, and $L$ is the total number of fully connected layers. The dimension of the last layer is different from the previous layers because the last layer is considered as an output layer, with $\mathbf{h}^L \in \mathcal{R}^k$, where $k$ is the size of gene sets in the gene expression matrix $\mathbf{X} \in \mathcal{R}^{N \times k}$.

## 3.3 Model Pre-training & Fine-tuning

**Pre-training.** The pre-training of *CellPLM* follows a cell language modeling objective, as demonstrated in Eq. (8). Specifically, given a batch of cell tokens as input, we first decide which cells should be masked. Instead of completely masking these cell tokens, we selectively mask a certain percentage of the gene expressions within them. This allows the model to recover underlying correlations between cells, as proposed in a recent preprint, SpaFormer [16]. A significant concern in *CellPLM* is the disparity in the number of genes measured by different sequencing platforms. Notably, the gap between scRNA-seq and SRT can be substantial, ranging from 1,000 to 30,000. Taking this into consideration, *CellPLM* only masks the expression of genes that are measured in each dataset, implying that the reconstruction loss is calculated exclusively on these measured genes. When optimizing the denoising variational lower bound in Eq. (8), we apply reparameterization trick and Monte Calo sampling, as proposed in VAE [37]. Furthermore, under the independent Gaussian assumption, we reformulate and estimate the reconstruction term $\mathcal{L}_{\text{recon}}$ in Eq. (8) with a mean squared error (MSE). Therefore, the pre-training loss function of *CellPLM* can be formulated as:

$$\mathcal{L}_{\text{MSE}} = \left\| \mathbf{M} \odot \left( \mathbf{H}^{(L)} - (1 - \mathbf{M}) \odot \mathbf{X} \right) \right\|_F^2, \quad \mathcal{L}_{\text{pretrain}} = \mathcal{L}_{\text{MSE}} + \mathcal{L}_{\text{cond}} + \mathcal{L}_{\text{Y}},$$
(9)

where $\odot$ signifies element-wise multiplication, $\mathbf{H}^{(L)} \in \mathcal{R}^{N \times k}$ is the output from the decoder, $\mathbf{X}$ and $\mathbf{M}$ are the ground-truth gene expression matrix and the mask indicator matrix respectively, as defined above. $\mathcal{L}_{\text{cond}}$ and $\mathcal{L}_{\text{Y}}$ are derived from Eq. (8).

**Task-specific Fine-tuning**. When fine-tuning *CellPLM*, the model is first initialized with the pre-trained parameters. In downstream tasks that require gene expressions as output, the pre-trained decoder is fine-tuned on the downstream datasets. Otherwise, the decoder will be replaced with a task-specific head. The entire model is then fine-tuned with task-specific loss functions, which helps align the general knowledge of the model to the specific downstream task. For example, in the spatial transcriptomic imputation task, the model is fine-tuned on a query SRT dataset and a reference scRNA-seq dataset, where two datasets are sampled from the same type of tissue. In this case, the loss function remains the same as Eq.(9). After fine-tuned on these datasets, *CellPLM* fit the data distribution of the target tissue and can readily perform imputation. The design and implementation of heads and loss functions for some downstream tasks are elucidated in Appendix E.

## 4 Experiment

*CellPLM* is first pre-trained on more than 9 Million scRNA-seq cells and 2 Million SRT cells, with the masked language modeling objective demonstrated in Section 3.3. To explore an appropriate model size, we created three different sizes of pre-trained models, with 5M, 10M and 40M parameters, respectively. All experiments were finished within 24 hours on a GPU server with 8 Nvidia Tesla v100 16GB cards. The hyperparameters, datasets, and reproduciability information for pre-trained models are detailed in Appendix D. Our preliminary results (See Appendix D) show that the 10M model achieved the best parameter efficiency. Therefore, in the downstream evaluation, we take *CellPLM* 10M as the base model without special mentioning.

In the following sections, we evaluate the performance of *CellPLM* 10M on various downstream tasks, including scRNA-seq denoising, spatial transctiptomic imputation, and perturbation prediction. With the selected tasks, we aim to answer the following research questions:

**RQ1:** Does *CellPLM* present extraordinary denoising power compared to non-pretrained models?

**RQ2:** Does *CellPLM* succeed in jointly modeling scRNA-seq and SRT data, thus benefiting from both the spatial information of SRT and the abundant transcriptomic profiles of scRNA-seq?

**RQ3:** Although being trained on a cell language model beyond single cells, does *CellPLM* also perform well on gene-level task?

### 4.1 Task 1: scRNA-seq Denoising

Given that single-cell RNA-Seq protocols capture only a subset of the mRNA molecules within individual cells, the resulting measurements exhibit substantial technical noise [38]. Therefore, we consider denoising power as the most desired and essential power for a single-cell foundation model. The goal of the denoising task is to estimate the true expression level of each gene in each cell from a noisy observation. To assess the denoising efficacy of *CellPLM*, we conduct an evaluation on two single-cell RNA-Seq datasets, i.e., PBMC 5K and Jurkat from 10x Genomics [39]. Following the setting of scGNN [13] and scGNN2.0 [40], we apply a random flipping process to a subset of non-zero entries, transforming them into zeros in order to simulate the effects of dropout. In order to establish a performance benchmark for *CellPLM*, we conduct a comparative analysis with contemporary approaches, including DeepImpute [41], scGNN2.0 [40], SAVER [42], DCA [43], MAGIC [44] and scImpute [45], which are considered state-of-the-art methods in the field. We evaluate scRNA-seq denoising performance based on two popular regression metrics, i.e., Root Mean Square Error (RMSE) and Mean Absolute Error (MAE), to measure the degree of similarity between predicted gene expression and the actual ones. More details pertaining to these methods, the fine-tuning of *CellPLM*, and the evaluation metrics under the task of scRNA-seq denoising can be found in Appendix E.1.

It is evident that the fine-tuned *CellPLM* consistently exhibits superior performance compared to all baseline models on both datasets. Note that even under the zero-shot setting, *CellPLM* shows satisfactory results that surpass five baselines on the Jurkat dataset. These observations support that our proposed *CellPLM* outperforms the state-of-the-art denoising techniques, which answers the question of **RQ1**. This superiority can be attributed to the knowledge it acquires from unsupervised pre-training.

Table 1: (*Task 1*) The scRNA-seq denoising performance on the PBMC 5K and Jurkat datasets.

| Model | PBMC 5K | | Jurkat | |
|---|---|---|---|---|
| | RMSE ($\downarrow$) | MAE ($\downarrow$) | RMSE ($\downarrow$) | MAE ($\downarrow$) |
| DeepImpute | $1.168 \pm 0.018$ | $1.051 \pm 0.025$ | $0.786 \pm 0.006$ | $0.557 \pm 0.003$ |
| scGNN 2.0 | $1.376 \pm 0.015$ | $1.237 \pm 0.019$ | $1.001 \pm 0.016$ | $0.917 \pm 0.021$ |
| GraphSCI | $1.068 \pm 0.007$ | $0.924 \pm 0.009$ | $0.659 \pm 0.030$ | $0.481 \pm 0.024$ |
| SAVER | $0.884 \pm 0.001$ | $0.748 \pm 0.001$ | $0.569 \pm 0.001$ | $0.472 \pm 0.001$ |
| DCA | $0.775 \pm 0.002$ | $0.621 \pm 0.002$ | $0.423 \pm 0.001$ | $0.351 \pm 0.001$ |
| MAGIC | $0.793 \pm 0.001$ | $0.639 \pm 0.001$ | $0.424 \pm 0.001$ | $0.351 \pm 0.002$ |
| scImpute | $1.170 \pm 0.003$ | $1.002 \pm 0.001$ | $0.624 \pm 0.002$ | $0.529 \pm 0.001$ |
| *CellPLM* (Zero-shot) | $0.920$ | $0.754$ | $0.543$ | $0.448$ |
| *CellPLM* (Fine-tuned) | $\mathbf{0.657 \pm 0.002}$ | $\mathbf{0.485 \pm 0.001}$ | $\mathbf{0.421 \pm 0.002}$ | $\mathbf{0.336 \pm 0.001}$ |

## 4.2 Task 2: Spatial Transcriptomic Imputation

Spatially resolved transcriptomics has revolutionized single-cell analysis by incorporating physical locations along with gene expression, leading to exciting breakthroughs. However, due to the highly detailed spatial resolution, spatial transcriptomic data at the cellular level often encounter substantial missing values, which pose challenges in data analysis. To assess the potential benefits of the pre-trained model in the given task, we evaluate *CellPLM* on two spatial transcriptomic datasets at single-cell resolution, i.e., Lung2 and Liver2 [46]. Following the setting of baselines including SpaGE [47], stPlus [48], gimVI [49] and Tangram [50], we impute the unseen genes of the SRT dataset utilizing a scRNA-seq dataset as reference. We identify the testing gene set in SRT data by stratified sampling according to gene sparsity [51] and holdout those genes in fine-tuning stage. To evaluate the accuracy of spatial transcriptomic imputation, we employ Root Mean Square Error (RMSE), Pearson correlation coefficient (Corr), and cosine similarity (Cosine) to measure the degree of similarity between the predicted spatial gene expressions and the corresponding ground-truth expression values.

Remarkably, the fine-tuned *CellPLM* takes the lead in all three metrics on both datasets. In addition, the impressive zero-shot performance indicates that *CellPLM* can leverage pre-training information to impute the SRT data, effectively addressing the research question **RQ2**. For additional information regarding baselines, the fine-tuning of the *CellPLM*, and the evaluation metrics under this task, please refer to Appendix E.2.

Table 2: (*Task 2*) The results of spatial tanscriptomic imputation on the Lung2 and Liver2 datasets.

| Model | Lung2 | | | Liver2 | | |
|---|---|---|---|---|---|---|
| | RMSE ($\downarrow$) | Corr ($\uparrow$) | Cosine ($\uparrow$) | RMSE ($\downarrow$) | Corr ($\uparrow$) | Cosine ($\uparrow$) |
| SpaGE | $0.617 \pm 0.032$ | $0.227 \pm 0.011$ | $0.352 \pm 0.015$ | $0.656 \pm 0.012$ | $0.253 \pm 0.014$ | $0.376 \pm 0.005$ |
| stPlus | $0.678 \pm 0.038$ | $0.177 \pm 0.021$ | $0.360 \pm 0.014$ | $0.801 \pm 0.044$ | $0.224 \pm 0.010$ | $0.399 \pm 0.012$ |
| gimVI | $1.230 \pm 0.081$ | $0.130 \pm 0.010$ | $0.325 \pm 0.010$ | $1.596 \pm 0.551$ | $0.163 \pm 0.019$ | $0.338 \pm 0.010$ |
| Tangram | $1.259 \pm 0.193$ | $0.123 \pm 0.005$ | $0.285 \pm 0.008$ | $1.209 \pm 0.157$ | $0.168 \pm 0.024$ | $0.309 \pm 0.008$ |
| *CellPLM* (Zero-shot) | $0.620$ | $0.237$ | $0.395$ | $0.686$ | $0.228$ | $0.408$ |
| *CellPLM* (Fine-tuned) | $\mathbf{0.612 \pm 0.013}$ | $\mathbf{0.251 \pm 0.011}$ | $\mathbf{0.402 \pm 0.019}$ | $\mathbf{0.641 \pm 0.011}$ | $\mathbf{0.278 \pm 0.008}$ | $\mathbf{0.427 \pm 0.004}$ |

## 4.3 Task 3: Perturbation Prediction

The perturb-seq technology has been established to examine the gene expression response at the single-cell level when subjected to pooled perturbations [52]. By comparing the gene expression before and after perturbation, downstream analysis of differential expression (DE) enables the identification of genes that play a crucial role in disease progression. To assess the potential benefits of *CellPLM* in the given task, we conduct experiments to predict the expression value of genes after perturbation. Following the setting of GEARS [53], we partition the perturbations into training, validation, and test sets, ensuring that none of the test perturbations are encountered during the optimization process. Two perturbation datasets are employed for evaluation: (1) the Adamson Perturb-Seq dataset [54], consisting of 87 one-gene perturbations; and (2) the Norman Perturb-Seq dataset [55], containing 131 two-gene perturbations and 105 one-gene perturbations. To evaluate the performance of perturbation prediction, we employ Root Mean Square Error (RMSE) to measure the degree of similarity between the predicted gene expressions and the corresponding ground-truth expression values. In addition,

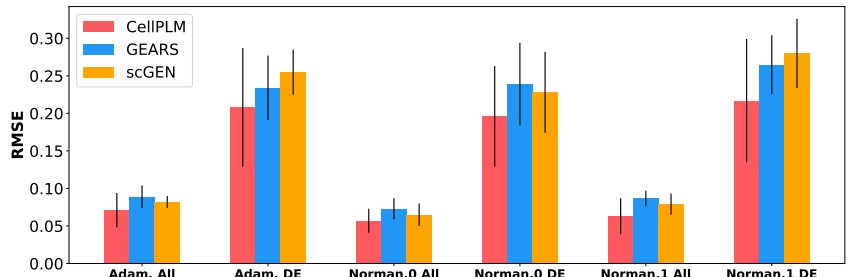

Figure 3: (*Task 3*) The RMSE performance (↓) on Adamson Perturb-Seq and the Norman Perturb-Seq datasets. The Norman Perturb-seq dataset consists of two settings: one-gene perturbations and two-gene perturbations, denoted as Norm.0 and Norm.1, respectively.

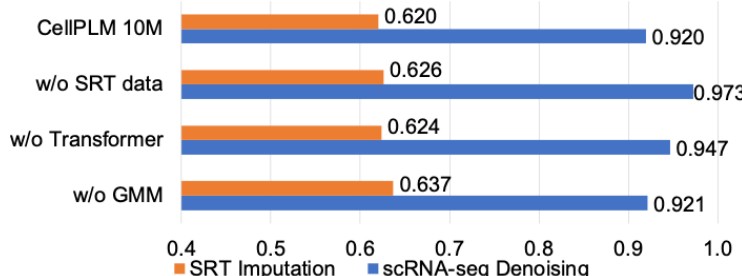

Figure 4: The ablation study of different pre-training settings. Zero-shot RMSE performance (↓) on PBMC 5K denoising task and Lung2 SRT imputation task, respectively.

following previous settings in GEARS [53], we also present the RMSE calculated on the top 20 deferentially-expressed genes.

We compare the performance between *CellPLM* and two baselines, i.e., a recent preprint GEARS method [53], and scGen [56]. The results in Figure 3 imply that *CellPLM* achieves the lowest RMSE values across all settings, which successfully tackles research question **RQ3**. For additional information regarding baselines, the fine-tuning of the *CellPLM*, and the evaluation metrics, please refer to Appendix E.3.

### 4.4    Ablation study

To verify the contribution of our model design, we conduct an ablation study on SRT data, Gaussian mixture prior and transformer encoder. Specifically, we remove SRT data from the pre-training dataset, replace transformer encoder with an MLP encoder and remove the Gaussian mixture prior, to examine its impact on the zero-shot performance in downstream tasks. All three models are modified based on *CellPLM* 10M. Our results demonstrate that, on the whole, the full 10M model exhibits the best performance, and its individual components display notable significance. Specifically, in the SRT imputation task, the GMM latent model contributes the most, while the removal of SRT data or the transformer component leads to the most substantial decrease in scRNA-seq denoising performance. The ablation study provides additional support, indicating that all elements within *CellPLM* offer valuable assistance in specific tasks.

## 5    Discussion

In this work, we propose cell language model, a novel paradigm of single-cell pre-trained model, which aligns well with the fundamental characteristics of single-cell data. This has leaded to *CellPLM*, the first pre-trained transformer framework that encodes inter-cell relations, leverages spatially-resolved transcriptomic data, and adopts a reasonable prior distribution. Our experiments on three downstream tasks demonstrate the power of *CellPLM*, which has a great potential to facilitate future research in single-cell biology.

**Limitations and future directions**: Despite the superior performance and results from the ablation study suggesting that our model has learned complex cell-cell relationships, extracting explicit knowledge and insights from the model remains a challenging task. Therefore, enhancing model interpretability is one foremost future objective. Moreover, due to the unavailability of implementations, we could not compare our model with existing pre-trained models. However, we intend to conduct a more comprehensive comparison in future studies.

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
