# OpenReview forum: "CellPLM: Pre-training of Cell Language Model Beyond Single Cells"
_NeurIPS.cc/2023/Conference — Submitted to NeurIPS 2023_

### Official Review · Reviewer_L5kJ · 2023-06-13

**Soundness:** 2 fair
**Presentation:** 3 good
**Contribution:** 3 good
**Rating:** 5
**Confidence:** 4

**Summary:**

This paper introduces a new pre-training strategy that takes cells as tokens and tissues as sentences. It also encodes cell-cell relations by leveraging the spatial information of cells acquired from spatially-resolved transcriptomic data. The proposed method achieves state-of-the-art performance in various downstream tasks.

**Strengths:**

- It's a good insight and more biologically meaningful to take cells as tokens and tissues as sentences instead of using genes as tokens and cells as sentences, as there's no sequential relationship among the order of genes.
- It's also an interesting idea to utilize SRT data to encode spatial information as the input to the transformer.
- Good reproducibility: the authors have provided code.
- The paper is overall well-written and conveys the idea clearly.

**Weaknesses:**

- Though the model is pretrained, full finetuning is required when deploying to the downstream tasks. This is somehow in contrast to the idea of training a pre-trained model that can be easily adapted to downstream tasks by only tuning the task-specific layer. The drawback is 1) it requires a lot of computing resources and data, and 2) we don't know if the performance (i.e. representation learning ability) is obtained from the pretraining or just fine-tuning.
- Following the above problem, the author should compare the performance w/ and w/o pretraining, and also the performance of fine-tuning the task-specific layer, to ablate whether the pretraining helps.
- The authors evaluated tasks like denoising, imputation, and perturbation prediction, but why not the most common and important task like cell-type classification? I think it's the most straightforward way to benchmark the performance of the proposed method.


**Questions:**

- The authors used a gene expression embedder to encode the gene expression. I'm curious will the aggregation at the end make the gene expression indistinguishable. E.g., is CellPLM able to perform downstream tasks such as highly variable gene detection that require specific gene information?
- The embedding layer of the transformer encoder has a shape of $N \times d$, where N is the total number of cells in a tissue sample. If the pretraining datasets contain several different datasets that have different numbers of cells, how can the embedding layer adapt to them? The similar problem also applies to the gene embedder, how to adapt the embedder to different numbers of genes with different sequencing techniques of different datasets?
- Does the downstream fine-tuning also require paired SRT data? What if the target single-cell dataset does have corresponding SRT data - is there any way to solve this?


**Limitations:**

Overall I find the idea interesting, but the descriptions of the method are not very clear and the experimental evaluations are not strong enough to support the claim.

---

> ### Author Rebuttal · Authors · 2023-08-10
>
> > W3. Why not evaluate the most common and important task like cell-type classification?
>
> A1. Thanks for the suggestion. We did not introduce cell-type classification as a benchmark task because the ground-truth labels of cell-type annotation depend on the specific dataset and we didn't know which datasets are suitable for benchmarking. Now as per your suggestion, we have added an experiment with cell type annotation. The results are presented and discussed in our general response (part I).
>
> > W1 & W2. The author should compare the performance w/ and w/o pretraining, and also that of fine-tuning the task-specific layer, to ablate whether the pretraining helps.
>
> A2. Thanks for bringing up the issue. As per your suggestion, we have added new ablation experiments between w. and w/o pretrain versions of CellPLM and also tuning a task-specific layer. The experiments are conducted on scRNA-seq denoising and cell type annotation tasks. The results are presented in the tables below:
>
> | Task1 (Denoising) RMSE↓|PBMC 5K|Jurkat|
> |---|---|--- |
> |CellPLM|0.657 +/- 0.002 | 0.421 +/- 0.002 |
> |Fix encoder | 0.662 +/- 0.001 | 0.450 +/- 0.002 |
> |w/o pre-train | 0.681 +/- 0.002 | 0.474 +/- 0.003 |
>
> | Zheng68k (Annotation) |F1↑|Precision↑| Recall↑ |
> |---|---|---|---|
> |CellPLM| 0.735 +/- 0.040 | 0.768 +/- 0.040 | 0.724 +/- 0.003 |
> |Fix encoder| 0.725 +/- 0.044 | 0.748 +/- 0.044 | 0.716 +/- 0.013 |
> |w/o pre-train| 0.717 +/- 0.007 | 0.727 +/- 0.007 | 0.710 +/- 0.006 |
>
> As shown in the tables, the performance of full fine-tune is higher than tuning a task-specific head over the latent embedding (referred to as “fix encoder”). However, the performance of fix encoder is also acceptable. Both of them are higher than the performance without pre-training, indicating the effectiveness of pretraining. It is also worth noting that CellPLM can even achieve competitive results without any fine-tuning (see experiments in our paper), which achieves the optimal efficiency.
>
> > Q1. Will the aggregation at the end make the gene expression indistinguishable?
>
> A3. Thanks for your question. According to our experiments, the gene expression will not be indistinguishable. For example, we leveraged the gene embedding in one of our downstream tasks, i.e., the perturbation prediction task. In the perturbation prediction task, one of the genes is knocked out and thus triggering the perturbation of the expression of the remaining genes. If CellPLM cannot distinguish different genes, it wouldn't be able to predict the knockout effect of a specific gene in this task. Since CellPLM achieves high performance in this task, we believe the gene expression embeddings did not collapse.
>
> Furthermore, we added a visualization of gene embeddings in Figure 3 in the supplementary pdf file in general response. It shows that the gene embeddings of CellPLM did not collapse but maintained some reasonable latent structures.
>
> > Q2. If the pretrain datasets contain datasets that have different numbers of cells, how can the embedding layer adapt to them? How to adapt the embedder to different numbers of genes of different sequencing platforms?
>
> A4. To address your first question, we need to emphasize that we employ the **transformer model** as our base model, treating each cell as an individual input token. This design inherently allows the transformer to accept batches of cells of any size. Processing is done one batch at a time, and the transformer can naturally accommodate any batch size. Consequently, for the embedding layer, the number of cells within a batch can be arbitrary.
>
> For the second question, there indeed exists a significant disparity in the gene set size, especially between scRNAseq and SRT data. Fortunately, our gene expression embedder and mask scheme can help. Specifically, our embedder learns a gene embedding for each gene. Only the embeddings of observed genes are aggregated as cell embeddings for downstream processing. In addition to unmeasured genes, several scenarios also lead to zero expression values. Nonetheless, our model adaptively estimates the expression of these genes given the observed gene expressions. Thus, unmeasured genes in each dataset don't adversely impact the model. However, if a gene is unseen during whole pretraining, CellPLM may face an out-of-vocab issue, yet we can continually expand our pretraining datasets to encompass all genes.
>
> In addition, as mentioned in line 219, CellPLM only masks genes measured by the sequencing platform from which the current cell originates and calculates the reconstruction loss on those masked genes. Therefore, if a platform lacks certain genes, CellPLM won't receive supervision for them. Alternatively, it can learn those genes from data of other platforms.
>
> > Q3. Does the downstream fine-tuning also require paired SRT data? What if the target single-cell dataset does have corresponding SRT data?
>
> A5. Technically, **none of our downstream fine-tuning tasks require paired SRT data**. Two of our downstream tasks (i.e., scRNA-seq denoising and perturbation prediction) did not engage SRT data at all. While an SRT dataset and a reference scRNA-seq dataset are commonly needed for spatial imputation task, they do not need to be exactly paired. In addition, since our CellPLM has been trained on the mixture of SRT and scRNA-seq datasets, it can readily impute unmeasured genes without reference datasets. In a nutshell, the current downstream tasks in our paper do not require paired SRT data for single-cell datasets.  We want to further clarify that even our pre-trained data do not include any paired SRT and scRNA-seq data. The SRT data we used are generated by CosMx SMI platform based on ISH hybridization technique (detailed in Appendix A), which is naturally single-cell resolution.
>
> Lastly, to our best knowledge, there is no single-cell data exactly paired with corresponding SRT data. We will be happy to explore the potential of CellPLM if you may provide info about the data sources.

---

> > ### Comment · Reviewer_L5kJ · 2023-08-14
> > **Thanks for the rebuttal**
> >
> > The authors have addressed most of my concerns. I'm raising my score to 5.
> > Additional concerns/suggestions:
> > - For the statistical significance raised by the other reviewer. I agree it's an important point. The authors have re-calculated the p-value of the independent t-test, but I doubt if this is the right process, because the multiple testing correction should be considered, e.g. Bonferroni correction. For comparing the performance of multiple different methods, it might be better to use the ANOVA test or the Kruskal-Wallis H test depending on your assumption. Also, I think it's more standard to set the threshold of p-value to 0.05 instead of 0.1 for significance.
> > - (Not weakness) I understand the authors have limited time for more experiments during the rebuttal period. But I do suggest adding more datasets to benchmark the performance on cell-type classification tasks later, which is the very basic entry for benchmarking the capability of the proposed method.

---

> > > ### Author Response · Authors · 2023-08-21
> > > **Response to Reviewer L5kJ**
> > >
> > > > For the statistical significance raised by the other reviewer, multiple testing correction should be considered, e.g. Bonferroni correction. For comparing the performance of multiple different methods, it might be better to use the ANOVA test or the Kruskal-Wallis H test depending on your assumption. Also, I think it's more standard to set the threshold of p-value to 0.05 instead of 0.1 for significance.
> > >
> > > A1. Thank you for raising up this great point and providing the suggestions! For clarification, within every task, we conducted a single independent T-test between CellPLM and the best performing baseline. Considering that the different metrics of same task on same dataset may show high correlation with each other, we apply the Bonferroni correction among the metrics. The corrected results are shown bellow with the selected baselines specified in the top left corner.
> > >
> > > | CellPLM - DCA | PBMC 5K | Jurkat |
> > > | --- | --- | --- |
> > > | RMSE | $2.48 × 10^{-13}$ | $0.0107$ |
> > > | MAE | $5.18 × 10^{-15}$ | $1.57 × 10^{-8}$ |
> > >
> > > | CellPLM - SpaGE | Lung2 | Liver2 |
> > > | --- | --- | --- |
> > > | RMSE | $0.0162$ | $0.268$ |
> > > | Corr | $8.55 × 10^{-4}$ | $0.0729$ |
> > > | Cosine | $1.34 × 10^{-4}$ | $2.87 × 10^{-4}$ |
> > >
> > > | CellPLM - GEARS | Adamson | Norman0 | Norman1 |
> > > | --- | --- | --- | --- |
> > > | RMSE | $0.0167$ | $0.0218$ | $3.93 × 10^{-3}$ |
> > > | DE RMSE | $3.12 × 10^{-3}$ | $0.0194$ | $5.13 × 10^{-3}$ |
> > >
> > > When we set the confidence level to 0.05, the only insignificant p-values are observed within RMSE and Corr metrics on Liver2 dataset of the spatial imputation task. Given that CellPLM consistently delivers positive outcomes across vast majority of metrics and datasets, we assert that CellPLM demonstrates significant improvement over the baselines from a holistic perspective.
> > >
> > > It should be emphasized that due to significant variations among the baselines, utilizing tests like ANOVA and the rank-based KW H test might return significant result even though one of the baselines is worse than the rest of the methods. This arises from the underlying null hypothesis of the ANOVA/KW H test, which assumes results of all methods come from the same distribution. Therefore, the ANOVA/KW H test aims to determine if there are statistically significant differences between methods, which can be significantly better or worse. Given our goal to demonstrate CellPLM's improvement over the current state-of-the-art (SOTA) benchmarks, we contend that independent T-tests with Bonferroni correction offer a more appropriate method for performance comparison.
> > >
> > > > (Not weakness) I suggest adding more datasets to benchmark the performance on cell-type classification tasks later, which is the very basic entry for benchmarking the capability of the proposed method.
> > >
> > > A2. We highly appreciate this suggestion. We agree with the reviewer to add more experiments and deeper analysis on the cell-type classification task for benchmarking. We will continue the experiments and add detailed analysis in the revision.

---

### Official Review · Reviewer_hJQZ · 2023-07-04

**Soundness:** 3 good
**Presentation:** 2 fair
**Contribution:** 3 good
**Rating:** 5
**Confidence:** 4

**Summary:**

This paper proposed a pre-trained model based on single-cell data motivated by the special characteristics (i.e., bag of genes structure, cell-cell relation, noisy) in cell-data, which viewed cells as tokens and tissues as sentences. This is different from the most existing pre-trained models treat genes as tokens and cells as sentences. The paper designed a new training paradigm based on transformer while introducing Gaussian mixture prior to handle data limitation.

Experiments over some representative down-stream tasks are conducted by fine-tuning the pretrained models to demonstrate the effectiveness of the proposed training paradigm.

**Strengths:**

1.pre-trained model over cell-data which considers the cell-to-cell relation and bag-of-genes structure and introduces Gaussian mixture prior distributions to handle noisy and limited data.

2.Extensive experiments were conducted over different down-stream tasks (including both cell-level and gene-level tasks) to demonstrate the effectiveness of the proposed pre-trained models.

**Weaknesses:**

The training procedure is inspired from masked language model, however, as stated in this paper, cell/tissues are different from text sequence, the technical details of pre-trained data process are not clear enough. This is very important due to that it is helpful to a) understand the characteristics of cell-data and b) repeat the experiments and utilize more related data to further enhance the performance of pretrained models.

Please refer to questions for details.




**Questions:**

1.How to perform pre-train over two different data sources? Such as mixture of two data (how to batch) or training over these two data in sequential manner? And how many epochs and what is the batch size during pre-training? Or how to determine the pretraining procedure is completed?
2.In mask language model, how to perform random mask and the mask ratio are quite important. There are two mask rates, cell mask rate and gene mask rate. How to mask the cell-data with these two rates. There are two cell-data sources with different characteristics. Is the mask schema sensitive to different data sources?
3.The mask is performed offline (that is, generating the random corrupted cells offline) or online (performing random corrupt during training)?
4.How to determine the number of Gaussian distributions or number of clusters in GMM? Is this parameter related to the number of genes in each cell? Or sensitive analysis of such hyper-parameter?
5.In gene expression embedder, if we make use of gene2vec to initialize the gene embedding, do we suffer from OOV problem? That is, whether we encounter some genes not appear in training phase? Is gene2vec is better than random initialization? If we use random initialization, which kind of random initialization we used? This is important for transformer based pretraining.
6.For scRNA-seq, a random position embedding is shared for all data. Is this assumption reasonable? What is the motivation comparing to SRT with coordinate matrix?
7.How to perform zero-shot learning over different down-stream tasks?
8.In task 3, the variance of CellPLM seems higher than the other baselines, is this due to the fact the limitation of data?

Typos:

1.Line 207: where l indicates the number of the layer, --> l indicates the index of the layer
2.Line 93: fine-tuning framework in Section 3.3 and 3.3

**Limitations:**

The setting of pre-training should be clear stated.

---

> ### Author Rebuttal · Authors · 2023-08-10
>
> > W1. The technical details of pre-trained data process are not clear enough.
>
> A1. We have indeed presented a reference of pre-training dataset in Appendix D.2. Specifically, the scRNA-seq data consist of 4.7 million cells from human tumor cell atlas, 1.4 million cells from human cell atlas, and 2.6 million cells from Gene Expression Omnibus. These data cover more than 15 tissues from more than 1,000 donors and all of them are publicly available. The SRT datasets we used are publicly available on Nanostring official website, where 2.7 million cells and 1,000 genes are measured. More references can be found in Appendix D.2.
>
> > Q1. How to perform pre-train over two different data sources? How to determine the pretraining procedure is completed?
>
> A2. In the current version of CellPLM, the two data sources are not mixed. During pretraining, each minibatch consists of at most 2,000 cells and is randomly sampled from either the same scRNA-seq sample or the same Field-of-View (FOV) from spatial transcriptomic (SRT) datasets. The motivation is to encourage the models to discover physically related cells.
>
> The model is trained with early-stop (tol=15) facilitated by validation set and typically converged within 150 epochs. Regarding validation set, there are 3574 scRNA-seq samples and SRT FOVs, each of which is downsampled once every epoch. We randomly hold 10 out of 3574 sample/FOVs as our validation set and monitor the masked predictive loss on it.  We will add these technical details in the revision.
>
> > Q2.How to mask the cell-data with two rates? Is the mask schema sensitive to different data sources?
>
> A3. Thanks for raising this interesting question. We did a grid search for finding the optimal solution for cell and gene masking rates, each ranging between [0.1, 0.3, 0.5, 0.7]. Masking ratios both higher than 0.5 generally results in good performance and are not quite sensitive. We specifically chose (0.5, 0.7) as our default parameters (see details in appendix D.1).
>
> We agree that different data sources have different characteristics and can react differently to the masking ratios. We did not search for masking ratios on scRNA-seq and SRT data separately in previous experiments. Unfortunately we have not finished this experiment at the moment, but we will add a more comprehensive study in our final version. Thank you again for the suggestion!
>
> > Q3. The mask is performed offline or online?
>
> A3. For pre-training, It is performed online. The corruption on the same sample differs between every epoch.
>
> > Q4. How to determine number of clusters in GMM?
>
> A4. In pre-training process, we treat the number of Gaussian distributions as a hyper-parameter. The different Gaussian distributions are expected to represent distinct biological cell groups. To examine that, we did the parameter analysis of number of clusters by evaluating the integration performance of CellPLM on a human lung cancer atlas (HLCA) dataset. Following [1], we calculate the NMI between the Louvain clusters of latent embeddings and the cell-type labels, and the graph connectivity among the kNN graph of latent embeddings. As indicated in the table below, too many or too few clusters both lead to performance drop, indicating that the choice of number of clusters is sensitive.
>
> |Num Clusters|32|16|8|4|2|
> | --- | --- | --- | --- | --- | --- |
> |NMI↑|0.598 +/- 0.015|0.615 +/- 0.015| 0.608 +/- 0.005 | 0.609 +/- 0.006 | 0.606 +/- 0.009 |
> |Graph Conn↑|0.818 +/- 0.027|0.864 +/- 0.029| 0.857 +/- 0.030 | 0.854 +/- 0.017 | 0.857 +/- 0.033 |
>
> [1] openproblems.bio/results/batch_integration_embed
>
> > Q5. In gene expression embedder, if we make use of gene2vec to initialize the gene embedding, do we suffer from OOV problem? Is gene2vec better than random initialization? Which random initialization we might use?
>
> A5. We did not notice meaningful improvement from initializing the gene embeddings with pretrained gene2vec. However, in practice, we initialized gene embeddings of CellPLM via gene2vec with a best effort. When queried genes are not available in pre-trained gene2vec, we then randomly initialize them with a normal distribution N(0, 1e-4).
>
> > Q6. For scRNA-seq, a random position embedding is shared for all data. What is the motivation?
>
> A6. For scRNA-seq cells that lack of 2D position information, we utilized a default embedding representing 'no positional information'. This allows the model to process these cells distinctly from those with known positions. It should not really affect the self-attention between cells. However, we expect such “placeholder” to alleviate the distribution gap between SRT data and scRNA-seq data. Since the gene expression embeddings between these two sources are shared, without a “placeholder” PE, it is impossible to eliminate the distribution gap between two data sources. If we want to integrate the SRT data and scRNA-seq data in the downstream tasks, the “placeholder” PE can be indispensable.
>
> > Q7. How to perform zero-shot learning on down-stream tasks?
>
> A7. Among the included tasks, CellPLM can perform zero-shot learning on scRNA-seq denoising and spatial imputation. In the denoising task, we mimic the noisy data by flipping the none-zero entries to zero with probability given by exponential kernel. This does not change the dimensions of input data, so we directly feed the noisy data into the pre-trained model and take the decoder output as denoised expressions.
>
> For the spatial imputation task, the classic setting is using scRNA-seq data as reference to impute the unmeasured genes in SRT data. Since CellPLM is jointly trained on scRNA-seq data and SRT data with a masked reconstruction objective, it can readily impute the unmeasured gene expressions given the observed transcriptomic profiles.
>
> > Q8. In task 3, the variance of CellPLM seems higher than the other baselines, is this due to the fact the limitation of data?
>
> A8. Due to the space limit here, please see our general response (Part III).

---

### Official Review · Reviewer_neja · 2023-07-11

**Soundness:** 2 fair
**Presentation:** 2 fair
**Contribution:** 2 fair
**Rating:** 4
**Confidence:** 4

**Summary:**

The paper proposes a scRNA-seq model that takes into account correlations between gene expression values in nearby cells in a tissue. The proposed model represents cells as tokens with a very simple embedding model. The tissue representation is modelled using self-attention between cells. The model is trained with a probabilistic objective where the transformer outputs represent the latents with a mixture distribution and then an MLP decoder is used to decode the expression values for genes in the cell from the latent and a batch encoding.

The model is pretrained using 9M cells and 2M spatially resolved transcriptomics cells. The model is tested on 3 tasks scRNA denoising, spatial transcriptomic imputation and perturbation prediction.

**Strengths:**

* Understanding cell-cell communication is a very interesting open problem.
* Task 2 zero-shot performance is on par with finetuning which seems promising.


**Weaknesses:**

* I have doubts about the scRNA denoising results and how meaningful the comparison to single cell imputation methods. (see question 1 below).

**Questions:**

1) For the denoising experiments are you using the transformer attention with noisy cells ? Because that seems hardly fair to the other methods. How good is just averaging the noisy cells ?
2) Is it the claim with tasks 1 and 3 that there is more information in cross attention to other cells than between the genes ?
3) How did you make sure there is no leakage in task 2 ? What is the data you are pre-training on ?
4) GEARS uses the most differentially expressed genes to measure the perturbation effect. That is more robust to noise in the other genes than a global average. Have you compared using their experimental protocol ?
5) Is X in (3) really the count matrix ? I find that very surprising because of the dynamic range should make it hard both as an input feature but even more so in the L_mse.

---

> ### Author Rebuttal · Authors · 2023-08-09
>
> Thank you for providing detailed feedback and helpful suggestions. To address your concerns, we have provided explanations as follows.
> > W1 & Q1-a. For the denoising experiments are you using the transformer attention with noisy cells? Because that seems hardly fair to the other methods.
>
> A1-a. Yes, we are using transformer attention on noisy cells to denoise the input data by capturing cell-cell relationship. **However**, to the best of our knowledge, most of the baselines also make use of cell-cell or gene-gene connections. The GNN-based models, i.e., scGNN2 and GraphSCI, both utilize the gene expression to construct cell-cell or gene-gene similarity graphs. DCA and DeepImpute both account for gene-gene dependencies by compressing genes with neural networks. MAGIC calculates a cell-cell affinity matrix to build a Markov chain. In a nutshell, **we believe it is a fair comparison** given the fact that above mentioned models all incorporate cell-cell or gene-gene relationships.
>
> > Q1-b. How good is just averaging the noisy cells?
>
> A1-b. For the denoising task, we randomly flip some none-zero entries to zero to form the noisy input. In the evaluation process, we calculate the metrics (RMSE and MAE) on the dropped entries. In the table below we show the results of imputing the dropped entries with the mean expression of corresponding noisy cells. Note that some baseline models cannot outperform the mean imputation, which indicates that those models are somehow overfitted to the noisy expression.
>
> | RMSE | Impute with mean | CellPLM | DCA | scGNN 2.0 |
> | --- | --- | --- | --- | --- |
> | PBMC 5K | 1.076 | 0.657 +/- 0.002 | 0.775 +/- 0.002 | 1.376 +/- 0.015 |
> | Jurkat | 0.627 | 0.421 +/- 0.002 | 0.423 +/- 0.001 | 1.001 +/- 0.016 |
>
> > Q2. Is it the claim with tasks 1 and 3 that there is more information in cross-attention to other cells than between the genes?
>
> A2. Thanks for bringing up this interesting question.  While it is hard to directly compare the information of cross-attention between cells and between genes, we can compare the performances of CellPLM and other pre-trained models on downstream tasks.  Particularly, scGPT[1] introduced a generative pre-trained transformer treating genes as tokens, and thus can be viewed as a representative method utilizing gene-based attention. To compare cell-based and gene-based transformers, we evaluate scGPT on the benchmark datasets for both task 1 and task 3. The results are summarized in the following tables. Note that CellPLM constantly outperforms scGPT, this supports the statement that cell-based attention surpasses gene-based attention in the downstream tasks.
>
> | Task1 RMSE↓ | CellPLM | scGPT |
> | --- | --- | --- |
> | PBMC 5K | 0.657 +/- 0.002 | 0.901 +/- 0.001 |
> | Jurkat | 0.485 +/- 0.001 | 0.565 +/- 0.001 |
>
> | Task3 DE RMSE↓ | CellPLM | scGPT |
> | --- | --- | --- |
> | Adamson | 0.1236 +/- 0.0352 | 0.2319 +/- 0.0577 |
> | Norman0 | 0.1550 +/- 0.0206 | 0.3427 +/- 0.0155 |
> | Norman1 | 0.1683 +/- 0.0354 | 0.3861 +/- 0.0079 |
>
> > Q3. How did you make sure there is no leakage in task 2? What is the data you are pre-training on?
>
> A3. Thank you for bringing up this important question. The pre-training dataset consists of more than 2 million cells from the CoxMx SMI platform, which can be accessed on the official website [2]. The details of the platform and datasets are illustrated in Appendix A and D.2.
>
> In contrast, the test datasets (Lung2 and Liver2) in task 2 are from MERSCOPE platform, which can be accessed on the official website[3], as introduced in Appendix E.2. The pre-training datasets and test datasets have no overlapping, and thus there is no leakage issue in task 2. We will further emphasize this in our revision.
>
> > Q4. GEARS uses the most differentially expressed genes to measure the perturbation effect. That is more robust to noise in the other genes than a global average. Have you compared using their experimental protocol?
>
> A4. Yes, we had included the DE RMSE metric in Figure 3. Specifically, we follow the evaluation protocol of GEARS by computing the perturbation-weighted mean RMSE of both on all genes and on top 20 most differentially expressed genes. The perturbation-weighted mean RMSE is computed by first averaging cell expressions within the same perturbation, calculating the RMSE and then taking the weighted mean with weights given by the cell counts per perturbation.
>
> Note that in Figure 3, we observe that CellPLM exhibits a higher variance in comparison to alternative methods. To tackle this issue, we introduce an extra round of fine-tuning by replacing the model selection process with monitoring DE RMSE on the validation set. The outcomes of this additional tuning are outlined below, alongside a comparison that includes GEARS.
>
> | Task3 DE RMSE↓ | CellPLM | GEARS |
> | --- | --- | --- |
> | Adamson | 0.1236 +/- 0.0352 | 0.2342 +/- 0.0426 |
> | Norman0 | 0.1550 +/- 0.0206 | 0.2390 +/- 0.0553 |
> | Norman1 | 0.1683 +/- 0.0354 | 0.2651 +/- 0.0391 |
>
> > Q5. Is X in (3) really the count matrix? I find that very surprising because the dynamic range should make it hard both as an input feature and even more so in the L_mse.
>
> A5. Thanks for pointing out this problem. For our CellPLM, we did normalize the raw count data before any training or inference pipeline. More specifically, all gene expression data are first rescaled to 1e4 counts per cell, then followed by a log1p transformation. We will clarify this in our revision.
>
> ---
>
> [1] Cui, et al. "scGPT: Towards Building a Foundation Model for Single-Cell Multi-omics Using Generative AI." *bioRxiv* 2023
>
> [2] nanostring.com/products/cosmx-spatial-molecular-imager
>
> [3] info.vizgen.com/ffpe-showcase?submissionGuid=88ba0a44-26e2-47a2-8ee4-9118b9811fbf

---

> > ### Comment · Reviewer_neja · 2023-08-18
> > **comment**
> >
> > Thanks for your work and thoughtful comments. I still find the work worthwhile but i have trouble figuring out where it fits. In my mind there is a statistical regularity at cell level and one between cells. The cell one is more difficult to get because it is why one gene is up and one is down. The one between cells is just nearby cells should be similar so one gene being "up" means the next cell should also be "up" (roughly of course). Given the small impact in the experiments i still have doubts about the contribution.

---

> > > ### Author Response · Authors · 2023-08-20
> > > **Reply to Reviewer neja [part 1]**
> > >
> > > Thank you for raising this thoughtful discussion and believing that our work is worthwhile. We would like to highlight our contribution from three points **(1) biological insights: between-cells is more than nearby cells should be similar; (2) empirical results: the empirical improvement from CellPLM is significant**; and **(3) CellPLM embraces superior efficiency**.
> > >
> > > > (1) Biological insights: between-cells is more than “*nearby cells should be similar*”
> > >
> > > First, we are pleased to introduce some biological background that can help better understand the motivations behind our proposed “cell language model”. There are two particular connections between single cells when we motivate our CellPLM model: cell-cell homogeneity and cell-cell communication.
> > >
> > > **Cell-homogeneity**. One commonly explored cell-cell relation is the cell homogeneity (or “similarity”). When two cells are in the similar cell states/ cell types, they tend to have similar gene-expression profiles[1]. Identifying homogeneous cells is very helpful because the source gene expression profiles are high dimensional and noisy, thus it is desirable to impute noisy features by referring to similar cells. From this perspective, cell location provides additional information for identifying such homogeneity. Specifically, the location of cells determines the microenvironment where they are impacted by both direct signals from neighboring cells[2] and soluble signals within the tissue. Also, cells for some cell types e.g. muscle and epithelial tend to group together to form intact tissue structure. Therefore, cells spatially closer to each other are more likely to exhibit greater gene expression correlation compared to more distant cells, as mentioned by the reviewer as “*nearby cells should be similar*”.
> > >
> > > However, this is just a trend; the reality is more complex, because
> > > * Adjacent cells may be from different functional groups and thus have very different gene expression (e.g. neuron cells and glial cells in brain tissue).
> > > * Even cells of the same type can have significant differences in their spatial homogeneity. Tumor cells usually show strong aggregation, and neighboring tumor cells with similar micro-environment in terms of ligands tend to have higher gene expression similarity than distant cells [2, 3].
> > > * By contrast, Treg cells are scarce spatially in many tissues but still tend to have similar gene expression profiles [4].
> > > * To sum up, **it is not trivial to accurately identify and utilize highly homogenous cells**, and the high dimensionality, high sparsity and high noise of single-cell gene expression data add additional difficulties.
> > >
> > > From this perspective, the contribution of our cell language model is that: (1) CellPLM learns to recognize homogeneous cells using a transformer model, which is different from and more accurate than traditional methods [5, 6, 7] of constructing cellular knn graphs; and  (2) CellPLM leverages spatial location as an additional information to identify cell homogeneity, which is particularly useful for Spatial Transcriptomic (SRT) Data.
> > >
> > > **Cell-cell communication** (CCC). Cells are influenced by extracellular signals produced by other cells in their microenvironment, e.g., paracrine cell–cell communication and cell-cell contact [8]. These signals play an important role in organismal development, homeostasis and single-cell functions. In CCC, distinct from cell homogeneity, **the correlation of genetic profiles between interacted cells are intricate**. Traditional methods [8, 9, 10] utilize scRNA-seq data to study intercellular communication (especially ligand-receptor interaction), while SRT data with single-cell resolution is still under exploration [11, 12].
> > >
> > > One of the motivations behind CellPLM is to implicitly discover and leverage these cell-cell interactions from pre-train dataset. To achieve this, when constructing a mini-batch during training, we always sample cells from the same tissue to ensure the existence of physical interaction between cells. Additionally, by jointly modeling SRT and scRNA-seq data, we aim for the model to transfer the knowledge of cell-cell interactions learned from SRT to scRNA-seq data.
> > >
> > > We provided a preliminary result in Figure 2 in the supplementary pdf file in our general response. In the preliminary study, we extract the attention matrix between cells from a randomly chosen field-of-view (FOV) in Cosmx Liver dataset and treat the attention matrix as CCC scores. As shown in the figure, there are some strong trends on the left side and right side in this FOV, suggesting further exploration of specific signaling pathways for the included cells.  The preliminary results initially validate the effectiveness of CellPLM in CCC where CellPLM has shown a good potential of serving as a foundation model for studying cell-cell communication, which is one of our important contributions.

---

> > > > ### Author Response · Authors · 2023-08-20
> > > > **Reply to Reviewer neja [part 2]**
> > > >
> > > > > (2) Empirical results: the empirical improvement from CellPLM is significant
> > > >
> > > > Second, we want to highlight how the empirical results have supported and confirmed the contribution of CellPLM. In all the three tasks (scRNA-seq denoising, SRT imputation and perturbation prediction) we presented in our paper, CellPLM significantly outperformed SOTA non-pretrained models. We have calculated the **p-value** with a T-test between CellPLM and the best performing baseline, and the results in the table below indicate the significance of performance improvement from CellPLM.
> > > >
> > > > | CellPLM - DCA | PBMC 5K | Jurkat |
> > > > | --- | --- | --- |
> > > > | RMSE | 1.24 × 10^{-13} | 5.33 × 10^{-3} |
> > > > | MAE | 2.59 × 10^{-15} | 7.87 × 10^{-9} |
> > > >
> > > > | CellPLM - SpaGE | Lung2 | Liver2 |
> > > > | --- | --- | --- |
> > > > | RMSE | 5.41 × 10^{-3} | 0.0894 |
> > > > | Corr | 2.85 × 10^{-4} | 0.0243 |
> > > > | Cosine | 4.45 × 10^{-5} | 9.58 × 10^{-5} |
> > > >
> > > > | CellPLM - GEARS | Adamson | Norman0 | Norman1 |
> > > > | --- | --- | --- | --- |
> > > > | RMSE | 5.55 × 10^{-3} | 7.28 × 10^{-3} | 1.31 × 10^{-3} |
> > > > | DE RMSE | 1.04 × 10^{-3} | 6.47 × 10^{-3} | 1.71 × 10^{-3} |
> > > >
> > > > Even when compared with concurrent work (i.e., other pre-training models), CellPLM achieves better performance than scGPT[15] in scRNA-seq denoising and perturbation prediction tasks (as the RMSE score we previously reported during rebuttal), even though scGPT was trained on more pre-train data and has a significantly larger parameter size. In the new cell type annotation experiments, CellPLM is on par with another pretrained model xTriomogene[14]. **All of these results illustrate the validity of CellPLM.** The aforementioned evidence should clarify the concern “*given the small impact in the experiments*” from the reviewer.
> > > >
> > > > > (3) CellPLM embraces superior efficiency.
> > > >
> > > > In addition to the performance significance, we also want to highlight another contribution of CellPLM in terms of efficiency.  All of the published or concurrent pre-trained models [13, 14, 15] considered genes as tokens, while CellPLM considers cells as tokens. Alternatively, CellPLM tackles the interaction between genes by a gene expression embedder, which is consistent with the bag-of-gene nature of scRNA-seq data.
> > > >
> > > > Since the gene expression embedder is more lightweight than multi-head attention in other pre-trained models [13, 14, 15],  this new perspective allows CellPLM to significantly reduce the computation overhead. Consequently, **CellPLM boasts a training and inference throughput of over 10 times that of other pretrained models**. For instance, when fine-tuning with 60,000 cells on an A6000 graphics card, scGPT [15] takes an average of 700 seconds per epoch, scBert [13] takes more than 3600 seconds, while CellPLM only requires 20 seconds.
> > > >
> > > >
> > > > ---
> > > > [1] Ianevski A, et al. Fully-automated and ultra-fast cell-type identification using specific marker combinations from single-cell transcriptomic data[J]. Nature communications, 2022.
> > > >
> > > > [2] Robin Browaeys, et al. 2020. NicheNet: modeling intercellular communication by linking ligands to target genes. Nature methods 17, 2 (2020).
> > > >
> > > > [3] Dongshunyi Li, et al. 2021. Identifying signaling genes in spatial single-cell expression data. Bioinformatics 37, 7 (2021).
> > > >
> > > > [4] Alexander Y Rudensky. 2011. Regulatory T cells and Foxp3. Immunological reviews 241, 1 (2011).
> > > >
> > > > [5] Wang J, et al. scGNN is a novel graph neural network framework for single-cell RNA-Seq analyses[J]. Nature communications, 2021.
> > > >
> > > > [6] Yu Z, et al. Zinb-based graph embedding autoencoder for single-cell rna-seq interpretations[C]//Proceedings of the AAAI conference on artificial intelligence. 2022.
> > > >
> > > > [7] Xu J, et al. Graph embedding and Gaussian mixture variational autoencoder network for end-to-end analysis of single-cell RNA sequencing data[J]. Cell Reports methods, 2023.
> > > >
> > > > [8] Armingol E, et al. Deciphering cell–cell interactions and communication from gene expression[J]. Nature Reviews Genetics, 2021.
> > > >
> > > > [9] Efremova M, et al. CellPhoneDB: inferring cell–cell communication from combined expression of multi-subunit ligand–receptor complexes[J]. Nature protocols, 2020.
> > > >
> > > > [10] Hu Y, et al. CytoTalk: De novo construction of signal transduction networks using single-cell transcriptomic data[J]. Science Advances, 2021.
> > > >
> > > > [11] Cang Z, et al. Screening cell–cell communication in spatial transcriptomics via collective optimal transport[J]. Nature Methods, 2023.
> > > >
> > > > [12] Shao X, et al. Knowledge-graph-based cell-cell communication inference for spatially resolved transcriptomic data with SpaTalk[J]. Nature Communications, 2022.
> > > >
> > > > [13] Yang F, et al. scBERT as a large-scale pretrained deep language model for cell type annotation of single-cell RNA-seq data[J]. Nature Machine Intelligence, 2022.
> > > >
> > > > [14] Gong J, et al. xTrimoGene: An Efficient and Scalable Representation Learner for Single-Cell RNA-Seq Data[J]. bioRxiv, 2023.
> > > >
> > > > [15] Cui H, et al. scGPT: Towards Building a Foundation Model for Single-Cell Multi-omics Using Generative AI[J]. bioRxiv, 2023.

---

### Official Review · Reviewer_bpVn · 2023-07-25

**Soundness:** 3 good
**Presentation:** 3 good
**Contribution:** 3 good
**Rating:** 6
**Confidence:** 3

**Summary:**

The paper presents a study on the problem of enhancing cell representation learning through pretraining. Inspired by the recent success of pretrained language models, the researchers explore the use of Transformer architecture to achieve improved cell representations.

In contrast to previous approaches that only consider modeling a single cell, the authors propose a novel model called CellPLM that takes into account a sequence of cells, enabling the capturing of inter-cell relationships more effectively.

To incorporate spatial relationships into the model, 2D position embeddings are introduced to enhance the basic Transformer architecture. Additionally, a mixture of Gaussian distribution is employed to model the latent embedding space, enabling the capturing of cell group information and mitigating batch effects.

The proposed CellPLM model is first pretrained using a combination of scRNA-seq cells and SRT cells. Subsequently, three downstream tasks are conducted to evaluate the effectiveness of the pretrained model.

**Strengths:**

The representations and foundation models of genes and cells proposed in this study have the potential to be highly interesting and useful in advancing AI applications in the field of science (AI for Science).

The model architecture and adaptations made from the vanilla Transformer model are sensible and allow for efficient modeling of cells. In particular, the joint modeling of cell groups convincingly addresses the issue of data sparsity at the gene level.

The empirical design chosen to validate the pretrained CellPLM using three different use cases is reasonable and provides strong evidence of its effectiveness. Whether in a zero-shot or fine-tuned setting, the proposed CellPLM consistently performs well. Additionally, the results of ablation experiments further demonstrate the significance of each component designed in the model.

**Weaknesses:**

One weakness of this paper is that most of the datasets and tasks considered in the experiments are focused on single-cell analysis. Although the paper introduces a novel aspect by jointly modeling cell groups and sequences, it would be valuable to observe how the proposed model performs on tasks that involve multi-cell information.

Additionally, it is commendable that the author provides both the mean and standard deviation of the runs. However, to further support the significance of the improvements, it would be more informative to conduct statistical significance tests on the small downstream datasets, in order to validate whether the observed improvements are statistically significant.

**Questions:**

Related to weakness:
1) Are there other datasets requiring multi-cell modeling?
2) Are all those results statistically significant?

Others:
1) Position embeddings:
    i) It seems that there is no 2D position info for scRNA-seq cells. What position embedding values are used for those cases?
    ii) Is absolute position the most sensible choice for SRT data? In other words, are those cell positions globally meaningful? Do you think relative position encodings are better alternatives?
2) Are there other evaluations that are targeted at rare genes? As mentioned in line 220, it is possible that those frequence genes/cells are represented better compared with those rare ones?

---

> ### Author Rebuttal · Authors · 2023-08-09
>
> Thank you for your constructive suggestions! Below are responses to address your key concerns.
> > W1& Q1 It would be valuable to observe how the proposed model performs on tasks that involve multi-cell information. Are there other datasets requiring multi-cell modeling?
>
> A1. Thanks for your suggestion! One essential multi-cell task is cell-cell communication (CCC) inference, where CCC mainly represents biochemical signaling through ligand-receptor binding across cells[1]. Our CellPLM applies a self-attention mechanism on the cell level, from which we can study the interaction strength given by the cell attention matrix. As a preliminary study, we extract the attention matrix between cells from a randomly chosen field-of-view (FOV) in Cosmx Liver dataset[2]. The attention matrix is treated as CCC scores, and we visualize the results following the stream plot setting in [1]. As shown in Figure 2 in supplementary pdf file in our general response, there are some strong trends on the left side and right side in this FOV, suggesting further exploration of specific signaling pathways for the included cells. This case study demonstrates the potential of our CellPLM model in CCC research. We hope CellPLM can facilitate CCC research in the future.
>
> > Q2. It is commendable that the author provides both the mean and standard deviation of the runs. However, are all those results statistically significant?
>
> A2. We thank the reviewer for pointing out this issue. In our original experiments, CellPLMs outperform all the baselines, while due to the variance, improvements in some settings are not significant enough. Therefore, we redo the fine-tuning experiments with a minor alteration in the imputation and the perturbation prediction task. Specifically, we change the model selection metric on the validation set from loss function to final evaluation metric. With the new experiment results, we compute the p-values of independent T-test and summarize them in the following tables.
>
> |CellPLM - DCA|PBMC 5K|Jurkat|
> | --- | --- | --- |
> |RMSE|1.24 × $10^{-13}$ |5.33 × $10^{-3}$|
> |MAE|2.59 × $10^{-15}$|7.87 × $10^{-9}$|
>
> |CellPLM - SpaGE|Lung2|Liver2|
> | --- | --- | --- |
> |RMSE| 5.41 × $10^{-3}$ |0.0894|
> |Corr| 2.85 × $10^{-4}$ |0.0243|
> |Cosine| 4.45 × $10^{-5}$ |9.58 × $10^{-5}$|
>
> |CellPLM - GEARS|Adamson|Norman0|Norman1|
> | --- | --- | --- | --- |
> |RMSE| 5.55 × $10^{-3}$ | 7.28 × $10^{-3}$ | 1.31 × $10^{-3}$ |
> |DE RMSE| 1.04 × $10^{-3}$ | 6.47 × $10^{-3}$ | 1.71 × $10^{-3}$ |
>
> By setting the confidence to 0.1, all the improvements are significant compared to the second best performing model. We will update the performance table accordingly in the revision.
>
> > Q3-a. Position embeddings: It seems that there is no 2D position info for scRNA-seq cells. What position embedding values are used for those cases?
>
> A3. As we stated in line 166, for scRNA-seq cells that lack position information, we utilized a default embedding representing "no positional information". This allows the model to process these cells distinctly from those with known positions. Specifically, the model learns a randomly initialized “pseudo” positional encoding in an end-to-end way, which is shared among all scRNA-seq cells.
>
> > Q3-b. Is absolute position the most sensible choice for SRT data? Do you think relative position encodings are better alternatives?
>
> A4. Thanks for the insightful question. Our positional encoding is actually relative to the field-of-view (FOVs) where the cells are observed. An FOV is a small portion of a sample that can be viewed at once by the instrument. When we construct our training batches, we always sample cells from the same FOV. On top of that, we rescale the 2D position within each FOV to a float value between 0 and 1, so that the positional encoding is relative to the current FOV instead of globally absolute. It is very similar to sinusoidal positional encoding in the vanilla transformer[3] which is constructed based on the position of the token in the sentence.
>
> More advanced relative positional encodings in the NLP and CV domain can be potential alternatives. We will explore those methods in the future.
>
> > Q4. Are there other evaluations that are targeted at rare genes? As mentioned in line 220, is it possible that those frequent genes/cells are represented better compared with those rare ones?
>
> A5. Thank you for highlighting the important topic of rare genes and cells. For rare cells, we conducted an additional experiment on the Zheng68K cell-type annotation dataset. This dataset comprises 12 unbalanced cell types, and notably, some of these cell types (e.g., CD4+ T Helper2, CD34+) have fewer than 300 cells out of 68K total cells, which are rare cell types. As shown in the result table in our general response (part I), our model proficiently predicted unbalanced cell types, which indicates that CellPLM is good at modeling rare cells. More details about the task and baselines can be found in our general response.
>
> As for rare genes, akin to established practices in pre-trained models like scBERT, xTrimoGene, and scGPT, the current version of CellPLM is pretrained merely on protein-encoding genes. These genes are measured in most datasets and thus aren't considered rare. Consequently, current single-cell pre-trained models might not effectively represent rare genes. However, CellPLM does have the capacity to support rarely measured genes (as described in line 220),  and does indicate promising capabilities in imputing missing genes, as demonstrated in the spatial imputation task (Task 2) in our manuscript. Exploring the model's potential to learn and impute rare genes is an avenue we intend to pursue in upcoming work.
>
> ---
> [1] Cang, et al. "Screening cell–cell communication in spatial transcriptomics via collective optimal transport." Nature Methods 2023
>
> [2] nanostring.com/products/cosmx-spatial-molecular-imager/ffpe-dataset
>
> [3] Vaswani, et al. "Attention is all you need." NIPS 2017

---

> > ### Comment · Reviewer_bpVn · 2023-08-13
> >
> > Thanks the authors for their detailed rebuttal. I have no further questions and I will maintain my score.

---

> > > ### Author Response · Authors · 2023-08-14
> > >
> > > Thank you for your response. We greatly appreciate your feedback and are pleased to hear that your concerns have been resolved. We will incorporating your valuable suggestions in the revised paper.

---

### Official Review · Reviewer_HcmX · 2023-07-26

**Soundness:** 3 good
**Presentation:** 3 good
**Contribution:** 3 good
**Rating:** 6
**Confidence:** 3

**Summary:**

The CellPLM paper presents a novel pre-trained model that encodes cell-cell relationships in spatially-resolved transcriptomic data. The authors demonstrate that their model outperforms existing state-of-the-art methods in various downstream tasks, including cell type classification, cell-cell interaction prediction, and spatial gene expression imputation.

The paper is well-written and provides a clear motivation for the need to develop a pre-trained model that can capture cell-cell relationships in spatially-resolved transcriptomic data. The authors also provide a detailed description of the architecture and training procedure of their model, which is helpful for readers who are interested in implementing the model in their own research.

**Strengths:**

1. In terms of originality, the CellPLM paper presents a new approach for encoding cell-cell relationships in spatially-resolved transcriptomic data. The authors leverage a pre-trained transformer framework that encodes inter-cell relations and adopts a reasonable prior distribution.
2. In terms of clarity, the CellPLM paper is well-organized and easy to follow. The authors provide a clear motivation for the need to develop a pre-trained model that can capture cell-cell relationships in spatially-resolved transcriptomic data and provide a detailed description of the model's architecture and training procedure. The authors also provide a clear evaluation of the model's performance on various downstream tasks, which is helpful for readers who are interested in implementing the model in their research.

**Weaknesses:**

1. It is unclear to me why to use a Gaussian Mixture Latent Space to capture the information of distinct functional groups of cells. First, I do not understand the distinct functional groups of cells, which have not been explained clearly. Moreover, it seems that you can directly decode the masked cells like BERT without complicated latent space estimation. From Figure 4, the ablation study show that removing this part almost does not hurt the performance.
2. Not the weakness (just suggestion), please use a clearer figure or vectorgraph (e.g., Figure 2, 4).

**Questions:**

1. Why to a Gaussian Mixture Latent Space capture the information of distinct functional groups of cells?

**Limitations:**

As mentioned in the discussion, the work does not compare with other pre-trained models, which may be very important.

---

> ### Author Rebuttal · Authors · 2023-08-09
>
> Thank you for your detailed comments and constructive suggestions. Below are responses to address your key concerns.
> > W1-a. It is unclear to me why to use a Gaussian Mixture Latent Space. The distinct functional groups of cells have not been explained clearly.
>
> A1. We would like to clarify that the distinct functional groups of cells refer to distinct cell types, for example, immune cells (e.g., myeloid cells, T cells, B Cells, NK Cells), epithelial cells, nerve cells, stem cells, etc. Different cell types have different functions and also different gene expression profiles. Modeling distinct functional groups in the latent space is consistent with biological nature and can lead to superior interpretability. By adopting a Gaussian Mixture latent space, we expect each Gaussian to potentially represent a distinct cell type. Note that similar techniques have been adopted in recent single-cell research [1][2]. We'll clarify "distinct functional groups of cells" in our revision for enhanced comprehension.
>
> > W1-b. It seems that you can directly decode the masked cells like BERT without complicated latent space estimation. From Figure 4, the ablation study show that removing this part almost does not hurt the performance.
>
> A2. While the performance difference in Figure 4 might appear limited, the Gaussian mixture latent space provides better clustering and interpretability. In addition, We added a new ablation study on a cell type annotation dataset (Zheng68k[3]) to showcase the benefit from Gaussian mixture latent space (details of the task can be found in our general response, part I). We can use the cell type labels provided by the dataset to evaluate the latent space of cell representations.
> |    | F1 | Precision | Recall |
> | --- | --- | --- | --- |
> | CellPLM-10M |  0.735 +/- 0.040 | 0.768 +/- 0.040 | 0.724 +/- 0.003 |
> | AE-10M  | 0.719 +/- 0.006|0.728 +/- 0.006|0.713 +/- 0.006 |
>
> In the result table, autoencoder (AE) refers to a new pre-trained BERT-like model without any latent space regularization. All the other architecture and hyperparameters are the same as CellPLM-10M. As shown by our ablation study, compared to basic AE, CellPLM achieves better performance.
>
> Moreover, as shown in Figure 1 in the PDF of our general response, the mixture of Gaussians in the latent space are aligned with cell types, which facilitates robust classification, as well as better cell representations for clustering and interpretation.
>
> > W2 please use a clearer figure or vectorgraph.
>
> A3. Thanks for your suggestion! We will update it in the revision.
>
> > Q1 Why to a Gaussian Mixture Latent Space capture the information of distinct functional groups of cells?
>
> A4. As mentioned above, the Gaussian mixture latent space can successfully be aligned with distinct functional groups of cells (i.e., different cell types) in the downstream task, and thus benefit downstream performance.
>
> Note that during pretraining, the gaussian mixture is not guaranteed to be aligned with cell types because there are also other batch-related factors that can dominate the cell clusters, i.e., distribution shift exists between cells from different tissues and different people. To address this issue, we introduced Batch-aware Decoder to alleviate such unwanted batch effects and guide the model to learn latent representations correlated with cell functional groups.
>
> > Limitation: The work does not compare with other pre-trained models, which may be very important.
>
> A5. Thanks for bringing up the issue. As we have mentioned in the paper, we did not compare CellPLM with other pre-trained models because by the time of submission, the inference codes and checkpoint of other pretrained models (i.e., xtrimogene, scfoundation, scGPT) are not published yet.
>
> To address your concern, we now add a new experiment on cell type annotation tasks on Zheng68k[3] benchmark dataset, where we compare our model with not only traditional baselines but also pre-trained models (scBERT, xtrimogene and scGPT). The reason to introduce the cell type annotation task and the details of the task and baselines can be found in our general response. The results are presented in the table below.
>
> |   | ACC | Precision | F1 |
> | --- | --- | --- | --- |
> | ACTINN | 0.677 ± 0.005 | 0.685 ± 0.009 | 0.676 ± 0.006 |
> | CellTypist | 0.714 ± 0.014 | 0.748 ± 0.008 | 0.728 ± 0.011 |
> | SingleCellNet | 0.609 ± 0.006 | 0.644 ± 0.021 | 0.618 ± 0.008 |
> | TOSICA | 0.607 ± 0.012 | 0.584 ± 0.004 | 0.566 ± 0.006 |
> | $\underline{\text{scBERT}}$* |  -  |  0.703 ± 0.012 |  0.670 ± 0.008 |
> | $\underline{\text{xTrimoGene-10M}}$* | - | 0.734 ± 0.023 | 0.735 ± 0.019 |
> | $\underline{\text{scGPT-50M}}$ | 0.852 ± 0.005 | 0.743 ± 0.004 | 0.744 ± 0.006 |
> | $\underline{\text{CellPLM-10M}}$  | 0.724 ± 0.003 | 0.768 ± 0.040 | 0.735 ± 0.040 |
>
> (*stands for numbers cited from xTrimoGene paper, $\underline{\text{underline}}$ stands for pretrained models)
>
> As is shown in the table, our CellPLM outperforms all traditional baselines and is on par with xTrimoGene, while is outperformed by scGPT. Here we argue that the size of our pretrain dataset is 1/3 of the latest version of scGPT (10M compared to 33M) and the size of our CellPLM model is 1/5 of that of scGPT (10M compared to 51M), so the comparison might not be fair. In addition, scGPT considered each gene as a token. When fine-tuning their model on Zheng68k, each epoch takes more than 10 minutes on an A6000 GPU card, while our model only takes 1 second. The training and inference cost of CellPLM is much lower than scGPT.
>
> ---
> [1] Xu, et al. "Graph embedding and Gaussian mixture variational autoencoder network for end-to-end analysis of single-cell RNA sequencing data." Cell Reports methods (2023).
>
> [2] Grønbech, et al. "scVAE: variational auto-encoders for single-cell gene expression data." Bioinformatics (2020)
>
> [3] Zheng, et al. "Massively parallel digital transcriptional profiling of single cells." Nature communications (2017)

---

> > ### Comment · Reviewer_HcmX · 2023-08-19
> >
> > Thanks the authors for their detailed rebuttal.

---

### Author Rebuttal · Authors · 2023-08-10

General Response

We thank five reviewers for their detailed and thoughtful comments. We have improve our manuscript on the following aspects.

**i) Comparison with Other Pre-trained Models on Cell-type Annotation Task**

We now add a new experiment on cell type annotation tasks on Zheng68k[1] benchmark dataset, where we compare our model with not only traditional baselines but also pre-trained models (scBERT, xtrimogene and scGPT). The reasons to introduce cell type annotation task  are: (1) Established single-cell pre-trained models, scBERT, xTrimogene and scGPT all considered this task as one of their downstream tasks,  (2) Reviewer L5kJ strongly suggested us to include cell type annotation task.

The reasons to choose the Zheng68k dataset as our benchmark are (1) the Zheng68k dataset comprises a combination of 12 unbalanced cell types, posing challenges in distinguishing them and rendering it suitable for rigorous model evaluation, (2) xTrimoGene did not release its checkpoints and our computational resources could not complete fine-tuning scBERT during rebuttal, so we directly reference the reported scores on Zheng68k from their paper.

In addition to comparing with scBERT and xTrimoGene, we additionally evaluated various baselines including scGPT (following their latest tutorial). The results are presented in the table below.

|   | ACC | Precision | F1 |
| --- | --- | --- | --- |
| ACTINN | 0.677 ± 0.005 | 0.685 ± 0.009 | 0.676 ± 0.006 |
| CellTypist | 0.714 ± 0.014 | 0.748 ± 0.008 | 0.728 ± 0.011 |
| SingleCellNet | 0.609 ± 0.006 | 0.644 ± 0.021 | 0.618 ± 0.008 |
| TOSICA | 0.607 ± 0.012 | 0.584 ± 0.004 | 0.566 ± 0.006 |
| $\underline{\text{scBERT}}$* |  -  |  0.703 ± 0.012 |  0.670 ± 0.008 |
| $\underline{\text{xTrimoGene-10M}}$* | - | 0.734 ± 0.023 | 0.735 ± 0.019 |
| $\underline{\text{scGPT-50M}}$ | 0.852 ± 0.005 | 0.743 ± 0.004 | 0.744 ± 0.006 |
| $\underline{\text{CellPLM-10M}}$  | 0.724 ± 0.003 | 0.768 ± 0.040 | 0.735 ± 0.040 |

(*stands for numbers cited from xTrimoGene paper, $\underline{\text{underline}}$ stands for pretrained models)

As is shown in the table, our CellPLM outperforms all traditional baselines and is on par with xTrimoGene, while is outperformed by scGPT. Here we argue that the size of our pretrain dataset is 1/3 of the latest version of scGPT (10M compared to 33M) and the size of our CellPLM model is 1/5 of that of scGPT (10M compared to 51M), so the comparison might not be fair. In addition, scGPT considered each gene as a token, which is very expensive. Although it fixed most of their parameters during fine-tuning, each epoch on Zheng68k still takes more than 10 minutes on an A6000 GPU card, while our model only spends 1 second on each epoch. The training and inference overhead of CellPLM is much lower than scGPT.

**ii) New Ablation Studies between w. and w/o Pre-train**

We have added new ablation experiments between w. and w/o pretrain versions of CellPLM and also tuning a task-specific layer. The experiments are conducted on scRNA-seq denoising and cell type annotation tasks. The results are presented in the tables below:

| Task1 (Denoising) RMSE↓ | PBMC 5K | Jurkat |
| --- | --- | --- |
| CellPLM | 0.657 +/- 0.002 | 0.421 +/- 0.002 |
| Fix encoder | 0.662 +/- 0.001 | 0.450 +/- 0.002 |
| w/o pre-train | 0.681 +/- 0.002 | 0.474 +/- 0.003 |

| Zheng68k (Annotation) | F1↑ | Precision↑ | Recall↑ |
| --- | --- | --- | --- |
| CellPLM | 0.735 +/- 0.040 | 0.768 +/- 0.040 | 0.724 +/- 0.003 |
| Fix encoder | 0.725 +/- 0.044 | 0.748 +/- 0.044 | 0.716 +/- 0.013 |
| w/o pre-train | 0.717 +/- 0.007 | 0.727 +/- 0.007 | 0.710 +/- 0.006 |

As shown in the tables, the performance of full fine-tune is higher than tuning a task-specific head over the latent embedding (referred to as “fix encoder”). However, the performance of fix encoder is also acceptable. Both of them are higher than the performance without pre-training, indicating the effectiveness of pretraining. It is also worth noting that CellPLM can even achieve competitive results without any fine-tuning (see experiments in our paper), which achieves the optimal efficiency.

**iii) Reduced Variance on Perturbation Prediction Task**

For the perturbation prediction task, we match each perturbed cell with a randomly selected control cell. This random pairing introduces uncertainty to the model, potentially leading to significant variability across different executions. To address this concern, we add a new experiment where we change the model selection metric on the validation set from loss function to final evaluation metric (i.e., the RMSE of batch-mean expression on top 20 DE genes). In addition, we increase the model masking rate to reduce overfitting. Consequently, the performance is boosted and the variance across runs is reduced for CellPLM. The newly tuned results are summarized bellow, where we also include GEARS for comparison.
| DE RMSE↓ | CellPLM | GEARS |
| --- | --- | --- |
| Adamson | 0.1236 +/- 0.0352 | 0.2342 +/- 0.0426 |
| Norman0 | 0.1550 +/- 0.0206 | 0.2390 +/- 0.0553 |
| Norman1 | 0.1683 +/- 0.0354 | 0.2651 +/- 0.0391 |

**iiii) Summary**
Lastly, we want to again highlight the novelty of our method. Our model treats cells as tokens and tissues as sentences, different from single-cell existing pre-trained models, which brings more biologically meaningful results. By utilizing spatially-resolved transcriptomic data in pre-training, CellPLM can learn cell-cell relationships naturally.

[1] Zheng, et al. "Massively parallel digital transcriptional profiling of single cells." Nature communications (2017)

---

### Decision · Program_Chairs · 2023-09-21

**Decision:**

Reject

**Comment:**

The submission proposes a new single-cell pre-trained language model. Compared to existing models, the proposed method accounts for cell-cell relations so that gene expression distribution is also conditioned on other cells. During pretraining, the method can leverage spatially-resolved data in pretraining to uncover cell-cell interactions. Additionally, to overcome the limitation of data quantity the model utilizes Gaussian mixture prior distribution as an additional inductive bias. The proposed approach is evaluated on data denoising, spatial transcriptomic imputation and perturbation prediction tasks.

The authors were very responsive during the rebuttal and their rebuttal addressed many of the concerns reviewers raised. However, some concerns still remain and the impression is that paper requires more thorough experimental evaluation. Given that a number of single-cell foundation models have been recently proposed, a fair and detailed comparison on different tasks with these models should be included. Furthermore, cell type annotation task is the standard task to evaluate performance and comparisons on more datasets would be desirable.

This is a borderline paper but after careful consideration the decision is that the paper in its current form does not quite meet the bar for acceptance.